# Urban Surface Ozone Concentration in Mainland China during 2015–2020: Spatial Clustering and Temporal Dynamics

**DOI:** 10.3390/ijerph20053810

**Published:** 2023-02-21

**Authors:** Youru Yao, Kang Ma, Cheng He, Yong Zhang, Yuesheng Lin, Fengman Fang, Shiyin Li, Huan He

**Affiliations:** 1Key Laboratory of Earth Surface Processes and Regional Response in the Yangtze-Huaihe River Basin, School of Geography and Tourism, Anhui Normal University, Wuhu 241002, China; 2Helmholtz Zentrum München–German Research Center for Environmental Health (GmbH), Institute of Epidemiology, 85764 Neuherberg, Germany; 3Department of Geological Sciences, University of Alabama, Tuscaloosa, AL 35487, USA; 4School of Environment, Nanjing Normal University, Nanjing 210023, China

**Keywords:** ozone, clustering, dynamic variations, influencing factors, mainland China

## Abstract

Urban ozone (O_3_) pollution in the atmosphere has become increasingly prominent on a national scale in mainland China, although the atmospheric particulate matter pollution has been significantly reduced in recent years. The clustering and dynamic variation characteristics of the O_3_ concentrations in cities across the country, however, have not been accurately explored at relevant spatiotemporal scales. In this study, a standard deviational ellipse analysis and multiscale geographically weighted regression models were applied to explore the migration process and influencing factors of O_3_ pollution based on measured data from urban monitoring sites in mainland China. The results suggested that the urban O_3_ concentration in mainland China reached its peak in 2018, and the annual O_3_ concentration reached 157 ± 27 μg/m^3^ from 2015 to 2020. On the scale of the whole Chinese mainland, the distribution of O_3_ exhibited spatial dependence and aggregation. On the regional scale, the areas of high O_3_ concentrations were mainly concentrated in Beijing-Tianjin-Hebei, Shandong, Jiangsu, Henan, and other regions. In addition, the standard deviation ellipse of the urban O_3_ concentration covered the entire eastern part of mainland China. Overall, the geographic center of ozone pollution has a tendency to move to the south with the time variation. The interaction between sunshine hours and other factors (precipitation, NO_2_, DEM, SO_2_, PM_2.5_) significantly affected the variation of urban O_3_ concentration. In Southwest China, Northwest China, and Central China, the suppression effect of vegetation on local O_3_ was more obvious than that in other regions. Therefore, this study clarified for the first time the migration path of the gravity center of the urban O_3_ pollution and identified the key areas for the prevention and control of O_3_ pollution in mainland China.

## 1. Introduction

Ozone (O_3_), an important trace component of the natural atmosphere, is formed by the photochemical reaction of nitrogen oxides (NO_x_) and hydrocarbons in the atmosphere after being irradiated by ultraviolet light [1,2]. Studies have shown that long-term exposure to high concentrations of O_3_ in the atmosphere not only directly destroys the lung function and visual function of local residents, but also affects thyroid function and nervous and cardiovascular systems [3,4,5]. Many researchers have confirmed that O_3_ in the near-surface atmosphere poses a significant threat to the health of organisms. For example, a 10 ppb increase in 1-day ozone was associated with a 1.56% increase in emergency department and urgent care visits in the Washington, DC metropolitan area [6]. In the entire Northern Hemisphere, O_3_ pollution in the troposphere exhibits an increasing trend, especially in South and East Asia [7,8,9,10]. On the regional scale, the long-term dynamic evolution of near-ground O_3_ has not been systematically revealed. Therefore, it is necessary to research the dynamic evolution process and spatial distribution characteristics of near-ground O_3_ in order to control urban O_3_ pollution, and this goal motivates this study.

O_3_ pollution in the atmosphere has become increasingly prominent on a national scale in mainland China in the last two decades. Mainland China is not only one of the most densely populated areas in the world, but is also one of the fastest-growing areas of industrialization and urbanization. Studies have shown that in the past two decades, near-ground O_3_ pollution in various regions of mainland China has been increasing, and the health of more and more residents are threatened by O_3_ pollution [11,12,13,14]. Feng et al. concluded that the O_3_ concentration increased by 62.40% during particulate pollution episodes in Beijing (BTH) in the autumn from 2013 to 2015 [15]. Gao et al. analyzed the O_3_ Monitoring Instrument (OMI) satellite data and suggested that the O_3_ concentration in northeastern China rose from 81.3 to 87.4 μg/m^3^ during 2015–2018, and the nonattainment days kept increasing [16]. Xu et al. showed that under the influence of the southeast monsoon, O_3_ pollution in Shanghai became particularly serious when anthropogenic contaminants were mixed with biological volatile organic compounds [17]. Numerous studies have reported the level and distribution characteristics of O_3_ pollution in major metropolitan areas in mainland China. However, the spatial clustering and temporal dynamics of O_3_ concentration have not been systematically clarified for the entire mainland of China.

Cities are concentrated and sensitive areas of O_3_ pollution [18]. On the one hand, gaseous pollutants of NO_x_ and volatile organic compounds (VOC_S_), which are precursors of O_3_, mainly come from industrial activities and automobile exhaust [19,20]. On the other hand, ozone emitted in cities and suburbs concentrate inside cities along with urban heat island circulation, causing aggravation of ozone pollution in upper atmospheric layers [21,22]. In addition, both human and natural factors affect the migration behavior and dynamic variations of urban O_3_ pollution. Factors such as the atmospheric particulate matter concentration, gaseous pollutant concentration, urban traffic conditions, temperature inversion, wind speed, temperature, hours of sunlight, humidity etc. have significant effects on the formation and concentration levels of O_3_ [23,24,25]. However, most studies on the influence of factors on O_3_ pollution focused only on individual urban areas, while the influence of these factors on O_3_ has not been revealed on a national scale. In addition, hypothesis that the rugged terrain and variable vegetation of mainland China may have an impact on the migration and diffusion of O_3_.

Considering the above-mentioned reasons, this study explores the characteristics of spatial clustering and temporal dynamics of the O_3_ concentration in cities from mainland China on a long-term scale. We also distinguish the effects of topography and vegetation characteristics on the distribution of O_3_ in various regions of China. To achieve these goals, the rest of this work contains three subsequent steps: (1) the analysis of the spatiotemporal distribution of the urban O_3_ concentration in mainland China from 2015 to 2020; (2) the characterization of the clustering and dynamic evolution of the urban O_3_ concentration at both the national and regional scales; and (3) the clarification of the influence of different typical environmental factors on the O_3_ concentration in various regions in China.

## 2. Materials and Methods

### 2.1. Data Collection

The analysis of long-term and high-precision O_3_ concentration data helps to clarify the spatiotemporal distribution and evolution characteristics of urban O_3_ pollution in mainland China. We adopted the daily maximum 8-h average (MDA 8) O_3_ concentration from 1550 monitoring stations from 336 cities in mainland China spanning from 2015 to 2020. All monitoring sites were derived from urban sites, and non-urban air monitoring sites were excluded from this study. To control the validity of the data, we strictly followed the minimum requirements for statistical validity of O_3_ concentration data as stipulated in the Ambient Air Quality Standards (GB3095-2012) for statistical analysis. In addition, we have also taken into account the adverse effects of anomalous values at individual monitoring stations at a given moment in time. For this reason, we counted 8-h average concentration data for O_3_ containing the following two criteria: (1) at least 6 h of average concentration values were included; and (2) the difference between the individual moments of the same monitoring point does not exceed one times the minimum value over a period of six consecutive periods. Monitoring data that did not meet this requirement were excluded.

The O_3_ concentration data were collected from the national real-time city’s air quality release platform in the China National Environmental Monitoring Center (http://106.37.208.233:20035/ (accessed on 30 July 2021)). Meteorological factor data were derived from the National Meteorological Information Center (NMIC) of the China Meteorological Administration (http://data.cma.cn/ (accessed on 15 August 2021)), and the normalized vegetation index (NDVI) was obtained from the Resource and Environment Science Data Center of the Chinese Academy of Sciences (http://www.resdc.cn/ (accessed on 10 September 2021)). Detailed data types and sources are shown in Appendix A.

### 2.2. Methodology

#### 2.2.1. Spatial Autocorrelation Analysis

Spatial autocorrelation analysis is a method to explore the spatial distribution pattern, the degree of spatial dependence, and the potential spatial dependence of spatial variables at different locations. It is widely used in the research of air pollution [26,27]. In this study, the global spatial autocorrelation statistics (Moran’s index) were calculated to explore the spatial agglomeration characteristics of O_3_ distribution in cities in mainland China. Geoda 2.0 software (Chicago, IL, USA) was used to calculate the global Moran’s index, and after 999 Monte Carlo simulation tests, statistical tests were performed using the standardized statistic Z (test level *α* = 0.05). The local indications of spatial association (LISA) were used to identify the spatial concentration of O_3_ in a local area. The calculation and significance test formulas for global spatial autocorrelation and local spatial autocorrelation are shown in Equations (1)–(3):(1)$$I_{G\mathrm{lobal}}=\frac{n\sum_{i=1}^{n}\sum_{j=1}^{n}W_{ij}(x_{i}-\bar{x})(x_{j}-\bar{x})}{\sum_{i=1}^{n}\sum_{j=1}^{n}W_{ij}\sum_{i=1}^{n}{\left(x_{i}-\bar{x}\right)}^{2}}$$
(2)$$Z=\frac{I-E(I)}{\sqrt{VAR(I)}}$$
(3)$$I\mathrm{Local}=\frac{\left(x_{i}-\bar{x}\right)}{\sum_{i}{\left(x_{i}-\bar{x}\right)}^{2}}\sum_{j}W_{ij}(x_{i}-\bar{x})$$

In these equations, *I* is the Moran’s index; *n* is the number of cities; *x_i_* and *x_j_* are the O_3_ concentration of the *i*-th and *j*-th cities, respectively; $\;\bar{x}$ is the average of the O_3_ concentration in cities; *n* is the number of cities; *W_ij_* is the spatial weight matrix; *VAR*(*I*) represents the variance of the global Moran’s index; and *E*(*I*) represents the expected value of the Moran’s index.

#### 2.2.2. Hot Spot Analysis

Hot spot analysis was used to identify statistically significant spatial clustering characteristics of high values (representing hot spots) and low values (cold spots). The results (*z* score and *p* value) can be used to identify the spatial clustering locations of high- and low-valued elements. This method considers the characteristics of O_3_ in the neighboring areas. This method is of great significance in revealing the spatial clustering characteristics of atmospheric pollutants. Using this method, the Getis-Ord Gi* statistic was adopted to calculate each value of the O_3_ concentration. The Z score was used to test Gi*. Among them, if the *z* score > 1.96, it indicated that there was a high-value cluster or hot spot area of O_3_; if the *z* score < −1.96, it indicated that there was a low-value cluster or cold spot area of O_3_. The formula of Gi* is shown in Equations (4)–(6):(4)$$\bar{X}=\frac{\sum_{j=1}^{n}x_{j}}{n}$$
(5)$$S=\sqrt{\frac{\sum_{j=1}^{n}x_{j}{}^{2}}{n}}-{\left(\bar{X}\right)}^{2}$$
(6)$$Gi^{*}=\frac{\sum_{j=1}^{n}W_{ijxj}-\bar{X}\sum_{j=1}^{n}W_{ij}}{\sqrt[{s}]{\frac{[n\sum_{j=1}^{n}W^{2}{}_{ij}-(\sum_{j=1}^{n}W_{ij}{)}^{2}}{n-1}}}$$
where, *S* is the standard deviation of the O_3_ concentration, and the other parameters are the same as those in Equations (1)–(3).

#### 2.2.3. Standard Deviational Ellipse Analysis (SDE)

The function of SDE is to quantitatively describe geographic elements with the center, long axis, and short axis as the basic parameters [28]. The center of the SDE indicates the spatial distribution characteristics of the elements and their relative positions in space, and the long and short axes can reflect the spatial distribution of the elements. In this study, the spatial distribution and movement characteristics of O_3_ distribution in mainland China cities were revealed through the variation in the center of gravity, long axis, and short axis of the SDE. The formula of SDE is expressed by Equations (7)–(9).
(7)$${\bar{X}}_{w}=\frac{\sum_{i=1}w_{i}x_{i}}{\sum_{i=1}^{n}w_{i}}$$
(8)$$\sigma_{x}=\frac{\sqrt{\sum_{i=1}^{n}\left(w_{i}x_{i}\cos\theta -w_{i}\bar{y_{i}}\sin\theta \right)}}{\sum_{i=1}^{n}w_{i}^{2}}$$
(9)$$\sigma_{y}=\frac{\sqrt{\sum_{i=1}^{n}\left(w_{i}{\bar{x}}_{i}\sin\theta -w_{i}{\bar{y}}_{i}\cos\theta \right)}}{\sum_{i=1}^{n}w_{i}^{2}}$$

In the equations, (*x_i_*, *y_i_*) is the spatial region of the research object; *w_i_* is the weight; *i* is the decision-making unit; *x* and *y* represent the relative coordinates of each point from the center of the standard deviation ellipse, respectively; *θ* is the rotation angle of the distribution pattern; and *σ_x_* and *σ_y_* are the standard deviations of the *x*-axis and *y*-axis, respectively.

#### 2.2.4. Multiscale Geographically Weighted Regression Model (MGWR)

Compared with the classical geographic weighted regression model (GWR), the MGWR allows each variable to have a different spatial smoothing level, which can reduce the estimation error. The MGWR therefore is a further optimization and improvement of the GWR. It can optimize the specific bandwidths of each independent variable’s relationship with the dependent variable. Based on the preliminary estimation results of GWR, it can decide on an optimal influencing scale using the backward fitting technology to obtain a more accurate result [29]. The equation of MGWR is shown in Equation (10).
(10)$$y_{i}=\beta_{0}\left(u_{i},v_{i}\right)+\sum_{j=1}^{k}\beta_{bwj}\left(u_{i},v_{i}\right)x_{ij}+\varepsilon_{i}$$

In Equation (10), (*u_i_*, *v_i_*) are the coordinates of the local sampling location; *β_0_* is the bandwidth of intercept, and the subscript “bwk” denotes the bandwidth of variable k; *β_bwj_* is the regression coefficient of the *j*-th variable at *i*; and *ε_i_* is the random error at (*u_i_*, *v_i_*).

In this study, the GWR estimation is used as the initial estimation. Subsequently, the difference between the true value and the predicted value obtained by the initialization estimation is calculated, which represents the initialization residual ($\hat{\varepsilon }$) described by Equation (11):(11)$$\hat{\varepsilon }=y-\sum_{j=1}^{k}{\hat{f}}_{j}\;(f_{j}=\beta_{bwj}x_{j})$$

When the degree of change in the estimated value of the regression coefficient is less than 1 × 10^−5^, the model fitting is considered to be completed and the calculation stops. In this study, all variables were standardized before the model fitting. The calculation of the MGWR is based on the MGWR2.2 software (https://sgsup.asu.edu/SPARC (accessed on 25 September 2021)) developed by the Spatial Analysis Research Center (SPARC) of Arizona State University, USA, and the map production is completed by the ArcGIS10.6 software (Redlands, CA, USA).

#### 2.2.5. Multi-Factor Generalized Additive Model (MGAM)

MGAM is a free nonparametric regression model that can flexibly represent complex relationships between variables and freely detect the effects of nonlinear regression [30]. A smooth function is used to fit each feature in the model. Furthermore, the model automatically selects the appropriate polynomial to explain the relationship between the independent and dependent variables. The equation of MGAM is shown in Equation (12).
(12)$$y=\beta_{0}+f_{j}\left(K_{1}\right)+f_{j}\left(K_{2}\right)+f_{j}\left(K_{3}\right)+\cdots +f_{j}\left(K_{N}\right)$$

In Equation (12), *y* is the O_3_ concentration in the study area, *β*_0_ is the intercept of the model, *K*_1_, *K*_2_, *K*_3_… *K_N_* are the main environmental factors, and *f_j_* (*K*) is the smooth function of *K*_1_, *K*_2_, *K*_3_*… K_N_.*

## 3. Results and Discussion

### 3.1. Temporal Distribution Characteristics of O_3_ Concentration in Cities

#### 3.1.1. Interannual Variation of Urban O_3_ Concentration

The 90th percentile of MDA8 in mainland China in 2015–2020 is listed in Table 1. The O_3_ concentration of cities in mainland China exhibited an increasing trend from 2015 to 2018, and then turned into a decreasing trend after 2018. Among them, the urban O_3_ concentration increased by 6.84% from 2016 to 2017. The coefficient of variation of the urban O_3_ concentration in all years exceeded 10% (Table 1), indicating that the O_3_ concentration did not distribute uniformly at the monitoring sites, and that there might be characteristics of aggregation.

Furthermore, the O_3_ concentration in mainland China was higher than that of Europe, Iran, and India (Appendix A). In particular, the average frequency of high O_3_ concentration days (>160 µg/m^3^) in mainland China is significantly higher than that of the above-mentioned regions/countries. Studies have shown that the formation of urban O_3_ was due to the emission of NO_x_ and VOCs, which have undergone complex photochemical reactions in the air; on the other hand, the formation of O_3_ was induced by meteorological conditions such as high temperatures, strong solar radiation, and low relative humidity [31,32,33]. Compared with the above-mentioned countries/regions, the high urban O_3_ concentration in mainland China was due to the development of industry, the increase in the number of automobiles, and the rapid expansion of urban areas in the last two decades. The development of cities not only aggravated the emission of NO_x_ and VOCs [34], but also increased the temperature of the ground surface and the probability of temperature inversion in local areas [35].

#### 3.1.2. Monthly and Daily Variations of Urban O_3_ Concentration

The monthly and daily variations of the urban O_3_ concentration in mainland China from 2015 to 2020 are shown in Figure 1. On the monthly time scale, the O_3_ concentration in November, December, and January was lower than that in other months, while the O_3_ concentration in April, May, June, July, and August was higher than that in other months. The variations in the O_3_ concentration have obvious seasonal characteristics (summer > spring > autumn > winter). The strong exchange between the stratosphere and the troposphere led to increased O_3_ concentrations in spring and summer [36]. In addition, strong solar radiation generated a large number of hydroxyl radicals in the atmosphere, which stimulated the reaction of VOC and hydroxyl radicals to generate O_3_ [37,38]. Compared to summer, in autumn and winter the atmospheric troposphere was relatively stable, and precipitation decreased, which slowed down the dilution and diffusion of air pollutants. In addition, weaker solar radiation also slowed down the production of O_3_ in autumn and winter. However, the change of the O_3_ concentration in May should be the focus of attention: the period from May to June in 2017 and 2018 was the time with the highest O_3_ concentration in mainland China. On the daily time scale, the urban O_3_ concentration exceeded the ambient air quality standards of China on 17 May (162 µg/m^3^), 18 May (165 µg/m^3^), 27 May (179 µg/m^3^), 28 May (183 µg/m^3^), 29 May (165 µg/m^3^), 2017, and 23 May 2019 (162 µg/m^3^) (grade II 160 µg/m^3^) (GB 3095-2012). The aforementioned phenomenon suggested that the urban O_3_ concentration has clustering characteristics on the monthly and daily scales.

Two factors have contributed to the concentration of peak urban near-ground-level ozone concentrations in May and June. On the one hand, NO_x_ emissions from industrial agglomerations were elevated in May and June compared to other time periods [39]. On the other hand, the possibility of precursor phototransformation has been exacerbated by both the gradual increase in temperature and the increase in the duration of light.

### 3.2. Spatial Distribution Characteristics of Urban O_3_ Concentration

The trend analysis tool was used to perform a three-dimensional perspective analysis based on the urban O_3_ concentration and geographic spatial coordinate projection of mainland China (Figure 2). In the east-west direction (shown by the green line in Figure 2), the best-fit trendline of the urban O_3_ concentration presented a “J-shaped” curve, where the urban O_3_ concentration showed a trend of low in the west and high in the east. In the north-south direction (blue line in Figure 2), the best-fit curve presented an “inverted U-shaped” curve, where the urban O_3_ concentration in the mid-latitude regions of mainland China was higher than that in the low- and high-latitude regions. From a national perspective, the overlapping regions of the eastern coastal and mid-latitude regions (N30°~N40°) of mainland China were the key areas of urban O_3_ pollution from 2015 to 2020.

Based on the data of urban monitoring points, the detailed spatial distribution characteristics of the urban O_3_ concentration are plotted in Figure 3. From 2015 to 2016, the urban O_3_ concentration pollution area exhibited a spot-like distribution, mainly concentrated in the Beijing-Tianjin-Hebei region, Shandong Peninsula, and the Yangtze River Delta region (Figure 3a,b). However, by 2018, the polluted area expanded from a spot-like shape to a surface-like shape. The areas of high O_3_ concentration included most cities in North China, East China, and Central China, and even cities in Southwest China were also polluted by O_3_ (Figure 3c,d). From 2018 to 2020, the area of O_3_ pollution was significantly reduced, and it was confined to North China and a small part of the Yangtze River Delta (Figure 3e,f). For instance, since 2017, the near-ground O_3_ concentration in the Yangtze River Delta region has shown a downward trend, which was due to the concentrations of O_3_ precursors (NO_x_ and VOCs) emitted in 2017, which were at their peak during this time period. High concentrations of NO_x_ and VOCs directly contribute to elevated urban O_3_ concentrations [39]. The results showed that although the concentration of O_3_ in cities from mainland China has been at a high level, the degree and area of pollution are decreasing. The obvious regional aggregation characteristics of urban O_3_ pollution can also be found in Figure 3. The core areas of pollution were mainly concentrated in the North China Plain and the Yangtze River Delta. From 2015 to 2018, the expansion of the O_3_ pollution area in cities in China was affected by the reduction of atmospheric particulate matter (PM_2.5_) and NO_x_ emissions [40]. Studies revealed that the reduction of PM_2.5_ concentrations in the atmosphere slowed down the aerosol sink of hydroperoxyl-free radicals, thus promoting the production of O_3_ [41]. In addition, the increase in temperature and the decrease in humidity on the urban surface also exacerbated the increase in the urban O_3_ concentration [41]. From 2019 to 2020, the reduction in the extent of urban O_3_ pollution was due to the strengthened management and control of air pollutant emissions by the government and the impact of the COVID-19 epidemic [42,43].

### 3.3. The Clustering Characteristics of Urban O_3_ Concentration

#### 3.3.1. The Spatial Autocorrelation Characteristics of Urban O_3_ Concentration

From 2015 to 2020, the distribution of the urban O_3_ concentration in mainland China had significant spatial clustering characteristics (Table 2). The difference in the global Moran’s index of O_3_ distribution was statistically significant (*p* < 0.01, and Z > 2.58). The results indicated that urban O_3_ distribution from 2015 to 2020 had a significant spatial positive correlation. The probability of the O_3_ distribution showing randomness was less than 0.0001, and the clustering probability of the O_3_ distribution was much greater than the probability of random distribution (Table 2). Overall, the distribution of O_3_ has the characteristics of spatial dependence and aggregation. In addition, compared with the O_3_ distribution characteristics in other years, the spatial clustering characteristics of the O_3_ concentration in 2018 were the most obvious (Moran’s *I* = 0.7496).

A local spatial autocorrelation analysis (LISA) was used to analyze the spatial distribution characteristics of the O_3_ concentrations in different local area units in mainland China (Figure 4). The results showed that the spatial distribution of urban O_3_ presented the clustering characteristics of the same type in different regions (namely, the high-concentration area was adjacent to the high-concentration area (H-H), and the low-concentration area was adjacent to the low-concentration area (L-L)). Among them, the H-H distribution areas were mainly concentrated in Beijing-Tianjin-Hebei, Shandong, Jiangsu, Henan, and other regions. It has been confirmed that O_3_ pollution was exacerbated by an interaction of atmospheric pollutants (PM_10_, NO_2_, CO, SO_2_, and PM_2.5_) and meteorological elements (average wind speed, sunshine duration, evaporation, precipitation, and temperature) in metropolitan areas [25]. Therefore, there are three main reasons for the concentration of O_3_ in these areas. Firstly, these areas have a large population, rapid economic development, and a large number of motor vehicles, which have promoted a large amount of NO_x_ emissions. Secondly, compared with other regions in China, industrial enterprises such as petrochemicals, coal-fired power plants, and organic chemical industries gather in these places, resulting in more VOCs being emitted into the atmosphere [25,44]. Finally, the intense solar radiation and high temperatures in metropolitan areas in summer directly induce the photochemical behavior of O_3_; in addition, the urban thermal circulation retards the dispersion of atmospheric pollutants. The L-L area was mainly concentrated in the southwestern region (such as Tibet, Xinjiang, Yunnan, and Guizhou province) and the northeastern region of China (such as Heilongjiang and Jilin provinces). The reason for the low concentration of clustering in the southwest is that NO_x_ and VOCs emissions were less than in other regions. In the northeast, the emission time of O_3_ precursors was short and the photochemical reaction was weak, and hence most areas in this region exhibited the characteristics of low concentration clustering.

Hot spot analysis was used to further verify the accuracy of local spatial clustering of O_3_ (Appendix A). The results showed that the area of O_3_ clustering was wider than that calculated by LISA, and the clustering regions included North China, East China, and Central China. Moreover, from 2015 to 2020, the variations in the regions of urban O_3_ clustering were not obvious, and the clustering regions had been concentrated in the above-mentioned regions. Overall, the hot spot analysis showed that the clustering area of O_3_ was basically consistent with the results of the LISA analysis.

#### 3.3.2. Spatiotemporal Evolution Pattern of Urban O_3_ Concentration

To further investigate the characteristics of spatial clustering and temporal variation of the O_3_ concentration, a standard deviation ellipse analysis (SDE) was used (Figure 5). In general, the standard deviation ellipse of the urban O_3_ concentration covered the entire eastern part of mainland China, and the long axis of the ellipse showed a southwest-northeast trend. It is noteworthy that the direction of the long axis was basically parallel to the Chinese population boundary (Hu line), indicating that human activities played a leading role in the increasing O_3_ concentrations. In addition, parts of the southwest and northwest regions of China were also included in the ellipse. The occurrence of this phenomenon may be related to the emission of air pollutants in Western China and the diffusion of pollutants in the atmosphere.

From 2015 to 2020, the length of the long and short axes of the standard deviation ellipse changed, and the mean center also moved (Appendix A). The length of the long axis and the short axis had both shortened and then elongated from 2015 to 2020. The combination of Figure 4 and Appendix A shows that, from 2015 to 2017, the area of the standard deviation ellipse had shrunk from 3,944,147.97 km^2^ to 3,835,207.67 km^2^. Subsequently, from 2018 to 2020, the area of the standard deviation ellipse had shown an expanding trend from 3,848,261.55 km^2^ to 3,921,629.37 km^2^. The results indicated that urban O_3_ in mainland China presented a phenomenon of high concentration clustering and low concentration diffusion from 2015 to 2020. In addition, the geographic center of ozone pollution (mean center of the standard deviation ellipse) was in Nanyang City, Henan Province. The movement direction of the geographic center was first northeast and then southwest; among them, its maximum moving distance was 36.80 km from north to south and 27.84 km from east to west. Overall, the mean center tended to move to the south. This phenomenon implied that the reduction of the atmospheric O_3_ concentration in North China played an important role in the prevention and control of national O_3_ pollution. The emission control of O_3_ precursors in urban agglomerations in southern China (e.g., Yangtze River Delta, Pearl River Delta, etc.) are conducive to the reduction of urban O_3_ concentrations.

### 3.4. Factors Influencing the Distribution Characteristics of Urban O_3_ Concentration

After multiple covariance testing and PCA analysis, NO_2_, SO_2_, PM_2.5_, CO, DEM, temperature, sunshine hours, precipitation and NDVI were then respectively used as explanatory variables and were substituted into the OLS, GWR and MGWR model for fitting. Model fitting results show that the value of R^2^ fitted by the MGWR model was higher than that of the OLS and GWR models, and the AIC value of MGWR fitting was lower than the AIC value of OLS and GWR fitting. The results indicated that compared with OLS and GWR, the MGWR model was more suitable for analyzing the influencing factors of the O_3_ concentration in 2018 (Appendix A). By the calculation of MGWR, it was found that the regression coefficients of the NO_2_ concentration, the SO_2_ concentration, the PM_2.5_ concentration, Digital Elevation Model (DEM), sunshine hours, precipitation, and NDVI were significant overall. The regression coefficients of the other four variables (air pressure, wind speed, air temperature, and relative humidity) were not statistically significant. Among them, the concentration of SO_2_, PM_2.5_, DEM, sunshine hours, and precipitation were mainly positively correlated with the spatial distribution of O_3_, while NDVI were negatively correlated with the spatial distribution of O_3_ (Appendix A). On the spatial scale, emissions of atmospheric gas pollutants may contribute to urban O_3_ formation, and the precursors of secondary particulate matter in PM_2.5_, which also contains ozone precursors, can contribute to urban O_3_ concentrations. There is a certain relationship between the photochemical oxidation of SO_2_ and O_3_. A high O_3_ concentration greatly promotes the chemical conversion of SO_2_ to sulfate [25]. Meteorological influences play an important role in influencing troposphere ozone concentrations, but the dominant factors show different effects on ozone in different regions [45,46]. The duration of light and the amount of evaporation are related to solar radiation. Intense solar radiation (sunshine hours) drives active photochemical reactions, which increase urban ozone concentrations. Similarly, solar radiation is stronger at higher altitudes than at lower altitudes due to the weakening effect of clouds and water vapor at lower altitudes that exacerbate the solar radiation. In addition, the effect of vegetation on regional O_3_ pollution is also very closely related. Studies have shown that during drought, vegetation is less effective in removing O_3_ through stomata than during wet periods, exacerbating the extreme effects of O_3_ pollution [47].

Bandwidth directly reflected the spatial scale of different influencing factors in the spatial process [48,49]. The bandwidth value (BW) had a positive correlation with the spatial influence of the dependent variable. In other words, the larger the BW, the wider the influence range of the influencing factors on the independent variables; meanwhile, the spatial influence of the influence factor was similar and the spatial heterogeneity was smaller [46]. The NO_2_ concentration (BW 314), DEM (BW 298), and sunshine hours (BW 318) had a wide range of effects on the spatial distribution of O_3_, and the scope of their influence spread to the scale of the entire study area (Appendix A). The concentration of SO_2_ (BW 150), PM_2.5_ (BW 56), precipitation (BW 108), and NDVI (BW 132) had an impact on the spatial distribution of urban O_3_ on a local scale, and their influence had not spread to the entire study area.

Furthermore, the spatial pattern of different influencing factors on O_3_ was analyzed (Figure 6). In East China, South China, Central China, and North China, the NO_2_ concentration and the spatial distribution of O_3_ showed a positive correlation (Figure 6b). On the one hand, these places had always been the main population distribution areas in China. Strong human activities (e.g., vehicle emissions, fossil fuel combustion, and industrial emissions) caused the emission of NO_x_ and VOCs. On the other hand, these regions are in low- and mid-latitude regions in China, and radiation and light stimulated the formation of O_3_. Studies have shown that tropospheric O_3_ is a secondary pollutant produced by the photochemical reaction of NO_x_ and VOCs [50]. Therefore, solar radiation, NO_x_ concentration, and VOCs concentration directly determined the local O_3_ concentration. However, the correlation between the O_3_ concentration and the NO_2_ concentration in Northwest China, Northeast China, and Southwest China was not obvious, because the local emissions of VOCs and solar radiation were relatively small. The results showed that the variations of O_3_ in these areas were affected by other factors. In the southwest, northwest, and northeast regions of mainland China, urban SO_2_ concentration and O_3_ concentration showed a positive correlation (Figure 6c). The above-mentioned areas are the main coal-burning areas in mainland China, and the SO_2_ emitted during the coal-burning process is an important part of secondary inorganic aerosols [51]. However, in Eastern China, the relationship between O_3_ concentration and SO_2_ concentration was not obvious, which indicated that SO_2_ was not the main factor influencing urban O_3_ pollution in mainland China. Generally speaking, the concentration of PM_2.5_ directly affected the local O_3_ concentration in the entire mainland of China. The main component of PM_2.5_ was composed of gaseous pollutants (such as carbon, organic carbon compound, sulfate, nitrate etc.) [52]. Among them, a minority of the chemical composition of PM_2.5_ derived from direct emissions from pollution sources, and most of it originates from atmospheric photochemical transformations of SO_2_, NO_x_ and VOCs [53]. NO_x_ and VOCs were the precursors of O_3_ formation. Therefore, the O_3_ concentration and PM_2.5_ concentration had a mutual influence relationship, especially in North China, Central China, and Northwest China, where the local PM_2.5_ concentration and O_3_ concentration showed a clear positive correlation (Figure 6d).

On the time scale, the correlation between PM_2.5_ and O_3_ concentration in different seasons was further analyzed. Under the effect of different meteorological elements, the influence of PM_2.5_ concentration on O_3_ concentration was a complex process; however, urban PM_2.5_ and O_3_ concentrations showed a significant negative correlation in all seasons on the time scale. A significant negative correlation (*r* = −0.423) was found between O_3_ and PM_2.5_ in mainland Chinese cities from 2015–2020. Within the 99% confidence interval, the correlation coefficients between urban O_3_ and PM_2.5_ in different seasons were −0.293 (spring), −0.080 (summer), −0.272 (autumn) and −0.296 (winter). These phenomena have been attributed to the fact that high concentrations of particles lead to an increase in the optical thickness of the aerosol, which weakens the photochemical production rate of ozone and contributes to a decrease in ozone concentration. In addition, an increase in PM_2.5_ concentration could weaken atmospheric radiation and thus suppress ozone levels through the disappearance of UV light [45]. On an hourly scale, we found a particularly interesting finding that urban O_3_ concentrations are highly significantly negatively correlated with PM_2.5_ concentration at 1, 6, 12, 18, 24, 48 and 72 h in advance (Appendix A).

The influence of physical geographic elements on the O_3_ concentration suggested regional characteristics (Figure 6e–h). DEM had a significant positive correlation with the O_3_ concentration in Southwest and Northwest China. On the one hand, Southwest and Northwest regions of China are at high altitudes, and the solar radiation there was stronger than it was in other regions. On the other hand, the rugged terrain interfered with the diffusion and dilution of atmospheric pollutants. Solar radiation and temperature directly drove the photoreaction of atmospheric pollutants [54,55]. Therefore, for all cities in mainland China (especially in North China and Northwest China), when the sunshine time increased, the O_3_ concentration increased. The effect of precipitation on O_3_ was due to the large cloud fraction along with precipitation, which weakens the intensity of insolation and affects the photochemical production of O_3_. In addition, precipitation washes out O_3_ precursors and limits the O_3_ formation [56]. Vegetation removed air pollutants by preventing the migration and diffusion of atmospheric particles and plant transpiration. Especially in Southwest China, Northwest China, and Central China, the suppression effect of vegetation on local O_3_ was more obvious than that in other regions. However, in Northeast China and East China, the inhibitory effect of vegetation on O_3_ was not significant. The reason may be that the VOCs released by plants and the VOCs emitted by human activities increased the probability of local O_3_ generation.

### 3.5. Interaction of Different Factors on the Variation of Urban O_3_ Concentration

The variation of urban O_3_ concentration was the result of the compound action of multiple factors, rather than a simple single factor control [57]. To further explore the main driving factors of atmospheric pollutants and meteorological factors on the variation of urban ozone concentration on the Chinese mainland, NO_2_, SO_2_, PM_2.5_, DEM, SH, PREC and NDVI were selected as explanatory variables, and a GAM model under the interaction of multiple factors was constructed. Through pairwise combination, 21 groups of interaction terms were obtained. The results showed that these 21 groups of results all passed the significance test, and the effect on the change of ozone concentration was significant at the level of *p* < 0.01 (Appendix A). In the model (GAM), the adjusted *R*^2^ was 0.609, and the maximum variance explanation rate was 60.9%, indicating that each interaction term has a good explanation degree for O_3_ concentration, and the model fitting effect reached the standard. In addition, the interaction F value of SH and PREC (5988.0), SH and NO_2_ (5673.7), SH and DEM (5567.8), SH and SO_2_ (5522.0), SH and PM_2.5_ (5462.2) was higher than that of other factors, and the interaction between SH and other factors significantly affected the variation of O_3_ concentration. Numerous studies have confirmed the significant contribution of sunshine hours to urban O_3_ concentration, mainly because sunshine hours affected the intensity of solar radiation (ultraviolet light) and the temperature of photochemical reactions [57,58]. However, the interaction of sunshine hours and other factors on the change of O_3_ requires further detailed analysis.

The interaction between SH and NO_2_, SH and SO_2_, SH and DEM, SH and PREC, SH and NDVI all showed stimulating effects on the increase of O_3_ concentration (Figure 7a–f). This result indicated that the risk of urban ozone pollution was exacerbated by unfavorable conditions of both pollutants and meteorological elements in areas with strong solar radiation. In general, when the concentration of NO_2_ was below 100 μg/m^3^, the response of O_3_ concentration to the interaction of NO_2_ and other factors was not obvious. However, when the NO_2_ concentration exceeded the threshold (100 μg/m^3^), the O_3_ concentration showed a rapid increase as the concentration of NO_2,_ and other pollutants increased (Appendix A). PM_2.5_ concentrations showed different characteristics from other atmospheric pollutants. With the increase of PM_2.5_ concentration, the positive promotion effect of other factors on O_3_ was shielded (Appendix A). Some studies have confirmed that the O_3_ pollution season has been extended from winter to spring under the background of reduced PM_2.5_ and NO_x_ concentrations in the North China Plain [37]. However, it can be seen from the interaction relationship between PM_2.5_ and other factors that reducing the emission of PM_2.5_ containing sulfate, nitrate and organic components could significantly reduce the O_3_ concentration. The role of precipitation was similar to that of PM_2.5_, with high-intensity precipitation attenuating the contribution of other factors (Appendix A), but high-intensity precipitation and prolonged sunshine promoted the production of O_3_ (Figure 7e). When NDVI was lower than ~0.5, with the increase of NDVI and other pollutant concentrations, the O_3_ concentration showed an increasing trend. When NDVI exceeded ~0.5, even if the concentration of other pollutants increased, NDVI suggested the phenomenon of inhibiting the production of O_3_ (Appendix A). This is because under low vegetation coverage, the BVOCs released by vegetation can participate in photochemical reactions and promote the generation of O_3_, while under high vegetation coverage, the vegetation can block direct sunlight and inhibit the production of photochemical reactions [59,60]. In addition, VOCs (e.g., isoprene) released from plants contribute directly to ground-level O_3_ formation, especially under high temperature conditions. VOC emissions increase during warm seasons with high insolation and especially during the flowering periods, often exhibiting exponential temperature dependence [61,62]. Some of the VOCs (e.g., monoterpenes and sesquiterpenes) released by plants can cause an increase in near-surface particulate matter concentrations [63,64]. However, the transfer of VOCs between the gas and solid phases can slow or exacerbate the formation of urban O_3_ [61]. Under high temperature conditions, VOCs released by plants can promote the formation of O_3_ in cities. However, plants can also prevent the formation of O_3_ through surface adsorption and the weakening of light intensity. These two opposite behaviors have significant changes in different regions and times.

## 4. Conclusions

This study explored the spatial clustering and temporal dynamic evolution characteristics of the O_3_ concentration in cities from mainland China on a long-term scale. The different effects of topography and vegetation characteristics on the distribution of O_3_ in various regions of China were also verified. The results suggested that the O_3_ concentration in China’s cities showed an upward trend from 2015 to 2018, followed by a downward trend after 2018. The variations in the O_3_ concentration had obvious seasonal characteristics (summer > spring > autumn > winter), where the change of the O_3_ concentration in May should be the focus of attention. From 2015 to 2016, the urban O_3_ concentration pollution area exhibited a spot-like distribution mainly concentrated in the Beijing-Tianjin-Hebei region, the Shandong Peninsula, and the Yangtze River Delta region. However, by 2018, the polluted area expanded from spots to surface. The areas of high O_3_ concentration included most cities in North China, East China, and Central China, and even some cities in Southwest China were also polluted by O_3_. On the national scale, the distribution of O_3_ had the characteristics of spatial dependence and aggregation. On a regional scale, the O_3_ distribution areas of high concentration were mainly concentrated in Beijing-Tianjin-Hebei, Shandong, Jiangsu, Henan, and other regions. In addition, the standard deviation ellipse of the urban O_3_ concentration covered the entire eastern part of mainland China. Overall, the mean center of the standard deviation ellipse tended to move to the south with the time variation. DEM had a significant positive correlation with the O_3_ concentration in Southwest and Northwest China. Vegetation removed air pollutants by preventing the migration and diffusion of atmospheric particles and plant transpiration, especially in Southwest China, Northwest China, and Central China, where the suppression effect of vegetation on local O_3_ was more obvious than that in other regions.

## Figures and Tables

**Figure 1 ijerph-20-03810-f001:**
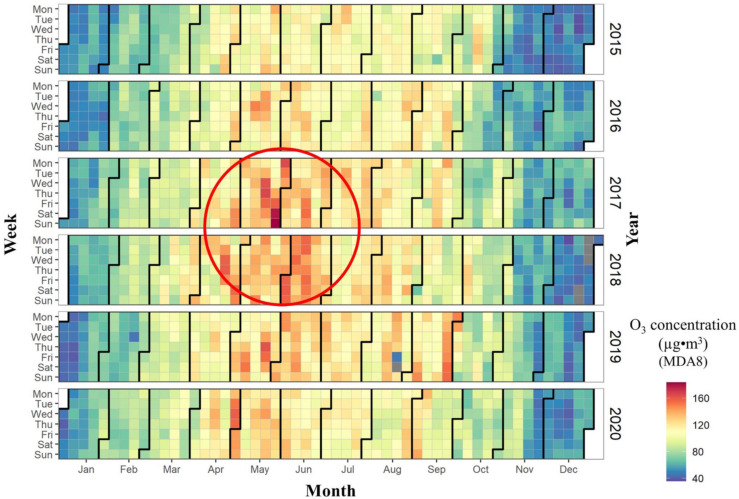
The heat map of urban O_3_ concentration (MDA8) in mainland China from the year of 2015 to 2020. (The “grey” colour in Figure 1 represents the days when there were less than 14 valid MDA8 data and the MDA8 value did not exceed the limit value criteria between 08:00 and 24:00. Missing dates are 22–26 December 2018).

**Figure 2 ijerph-20-03810-f002:**
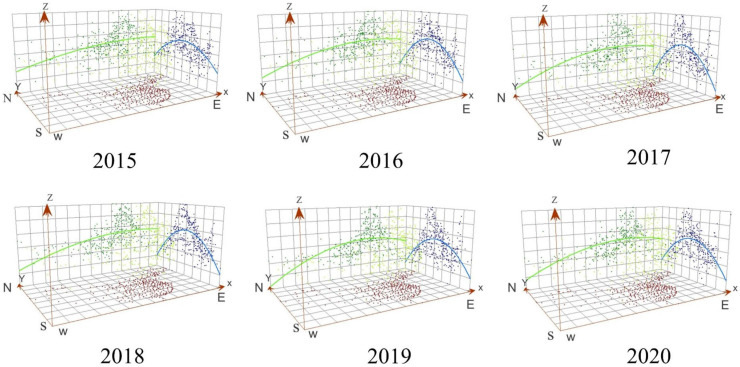
Trend surface analysis of the urban O_3_ concentration (the 90th percentile of MDA8) in mainland China from 2015 to 2020.

**Figure 3 ijerph-20-03810-f003:**
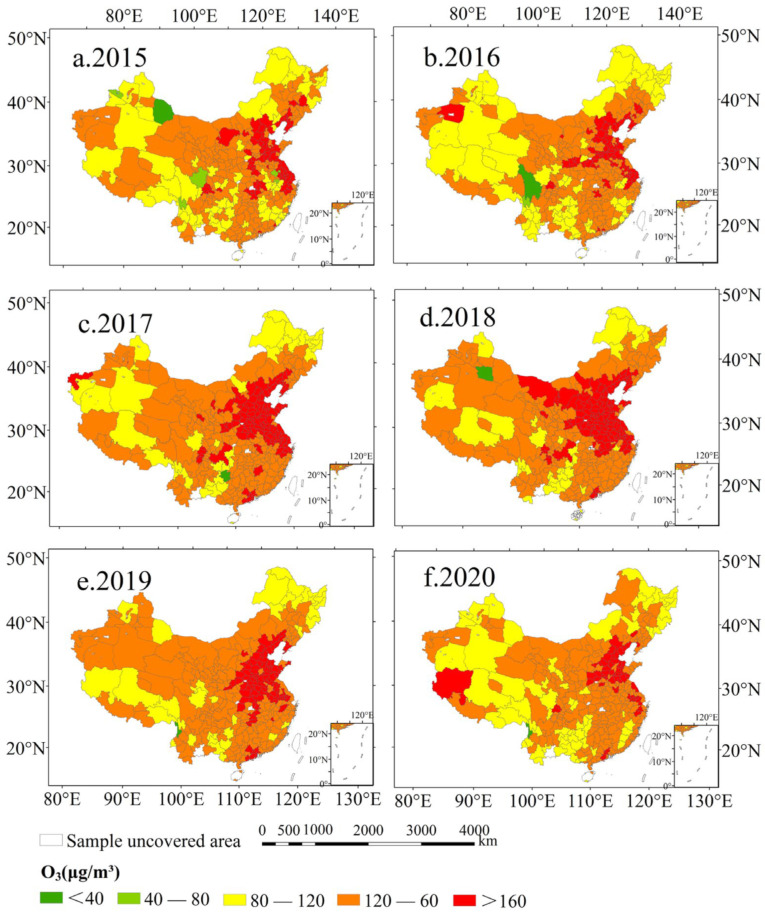
The spatial distribution of the urban O_3_ concentration (the 90th percentile of MDA8) in the mainland of China from the years 2015 to 2020. (**a**) 2015; (**b**) 2016; (**c**) 2017; (**d**) 2018; (**e**) 2019; (**f**) 2020. (Note: Urban O_3_ concentrations are derived from the average values of the effective monitoring stations in each city).

**Figure 4 ijerph-20-03810-f004:**
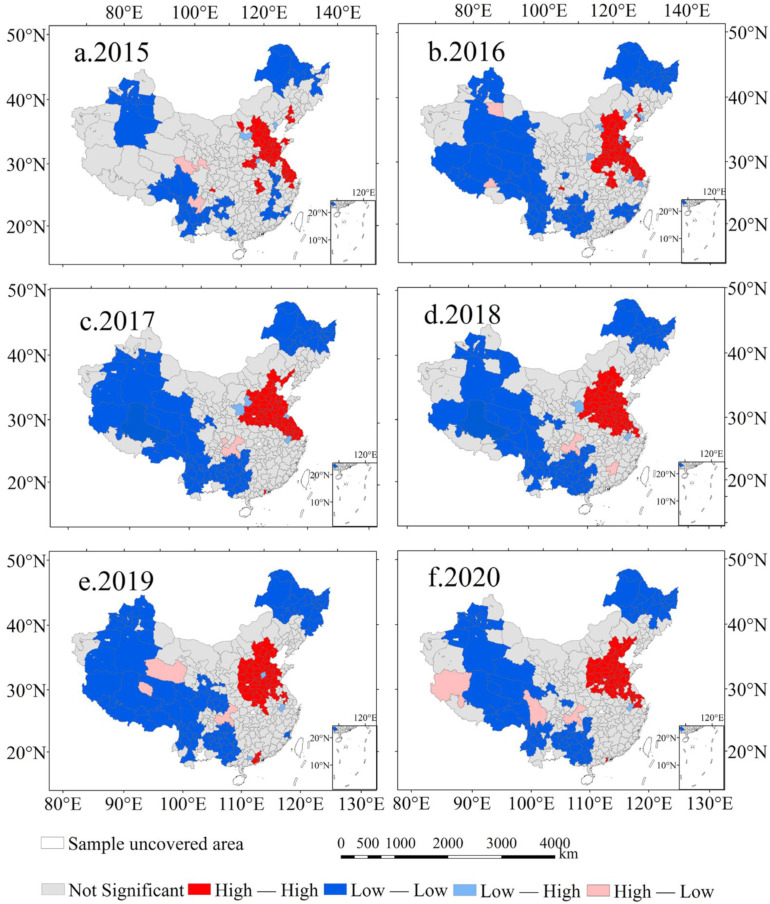
The spatial clustering pattern of the urban O_3_ concentration in mainland China from 2015 to 2020. (**a**) 2015; (**b**) 2016; (**c**) 2017; (**d**) 2018; (**e**) 2019; (**f**) 2020. (Note: Urban O_3_ concentrations are derived from the average values of the effective monitoring stations in each city).

**Figure 5 ijerph-20-03810-f005:**
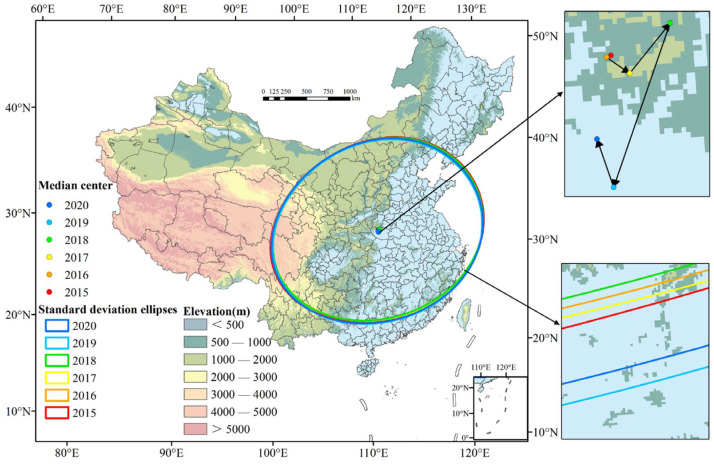
Time series of the mean center and standard deviational ellipse of the O_3_ concentration in mainland China in 2015–2020. (Note: Urban O_3_ concentrations are derived from the average values of the effective monitoring stations in each city).

**Figure 6 ijerph-20-03810-f006:**
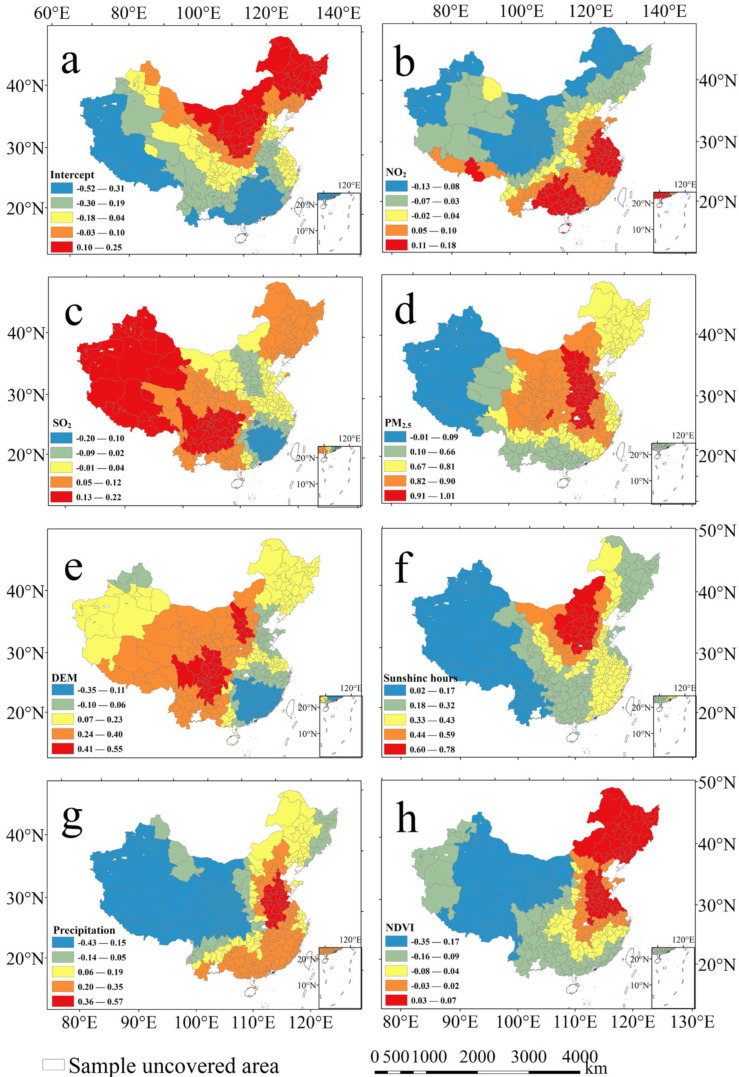
Spatial patterns of coefficients in the MGWR. (**a**) Intercept, (**b**) NO_2_, (**c**) SO_2_, (**d**) PM_2.5_, (**e**) DEM, (**f**) sunshine hours, (**g**) precipitation, (**h**) NDVI.

**Figure 7 ijerph-20-03810-f007:**
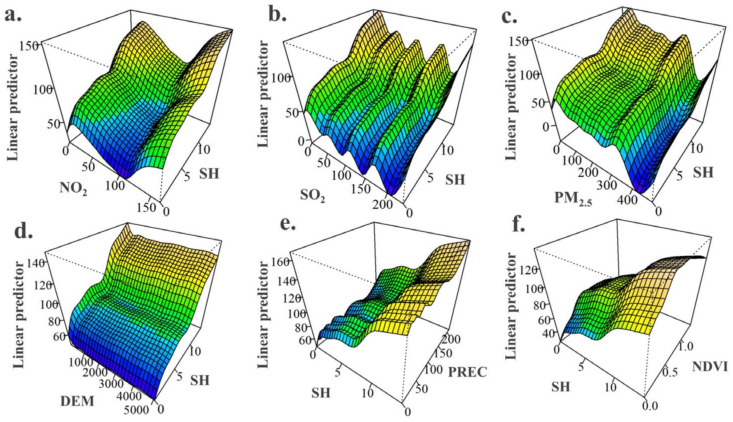
Interaction of different factors on the variation of urban O_3_ concentration Three-dimensional effect plots. (**a**) NO_2_ and sunshine hours (SH); (**b**) SO_2_ and sunshine hours (SH); (**c**) PM_2.5_ and sunshine hours (SH); (**d**) Digital Elevation Model (DEM) and sunshine hours (SH); (**e**) precipitation (PREC) and sunshine hours (SH); (**f**) normalized difference vegetation index (NDVI) and sunshine hours (SH).

**Table 1 ijerph-20-03810-t001:** Statistical data of the 90th percentile of the MDA8 from urban monitoring stations in mainland China (2015–2020) (µg/m^3^) ^#^.

Year	Counts	Min	Max	Mean	Coefficient of Variation (%)
2015	1491	42	227	142 ± 29	20.52
2016	1462	54	228	146 ± 27	18.52
2017	1425	35	236	156 ± 29	18.78
2018	1489	25	230	157 ± 27	16.92
2019	1579	32	225	152 ± 27	17.98
2020	1550	60	211	143 ± 24	17.10

^#^ MDA8, representing the annual mean O_3_ concentration of the maximum daily 8 h average.

**Table 2 ijerph-20-03810-t002:** Results of spatial autocorrelation analysis of the urban O_3_ concentration in mainland China.

Year	Moran’s I	E(I)	Z-Score	*p*-Value	Cluster	V(I)
2015	0.4908	−0.0030	14.1395	0.0001	Yes	0.0050
2016	0.6137	−0.0031	17.8051	0.0001	Yes	0.0032
2017	0.7496	−0.0030	21.5970	0.0001	Yes	0.0022
2018	0.7613	−0.0030	21.6733	0.0001	Yes	0.0021
2019	0.7538	−0.0030	21.7992	0.0001	Yes	0.0021
2020	0.7053	−0.0030	20.4490	0.0001	Yes	0.0024

## Data Availability

All data included in this study are available upon request by contact with the corresponding author.

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
