# Peer review of "Urban Surface Ozone Concentration in Mainland China during 2015–2020: Spatial Clustering and Temporal Dynamics"

_ijerph, 2023, doi:10.3390/ijerph20053810_

Round 1
Reviewer 1 Report
Multivariate analysis can be performed only after removing noisy components in the variables to avoid destructive effect of regressions as have been proven in the recent book "High resolution ...." by Cambridge Scholars.
Author Response
Dear Reviewer,
Thank you very much for handling our submission and the constructive comments from the referees for our manuscript No. ijerph-2146763.
All the questions raised by the reviewers were addressed in the revised manuscript with highlights.
- Multivariate analysis can be performed only after removing noisy components in the variables to avoid destructive effect of regressions as have been proven in the recent book "High resolution ...." by Cambridge Scholars.
Response: Thanks for the reviewer's comment. When two or more explanatory variables are highly correlated in the MGWR model, the results are subject to multicollinearity(Breiman et al, 1985; Brunsdon et al, 1996). To avoid bias and uncertainty in the model output results that may be caused by multiple co-linearities, we have identified and prevented each of the three possible cases of multiple co-linearities in the MGWR model. The specific treatment process is as follows:
- One of the explanatory variables is spatial clustering. To prevent this, we mapped each explanatory variable and identified variables that might have few values or where the same variables were spatially clustered(Brunsdon et al, 2017). If we observe these types of variables, we consider removing them from the model or representing them in a way that increases the range of values. However, We did not find explanatory variables of this type in this study.
- Two or more explanatory variables are highly correlated at the global level. For this case, we use a generalised linear regression model (GLM) to examine the variance inflation factor (VIF) of each explanatory variable. If the VIF value is greater than 7.5 or higher, global multicollinearity may prevent MGWR from running(Oshan et al, 2019). In this case, some explanatory variables are redundant, so we considered removing explanatory variables with VIF values >7.5 (PM10 and Air pressure) from the model. Finally, the explanatory variables retained were PM2.5, CO, NO2, SO2, Temperature, Wind speed, Relative humidity, Sunshine hours, Precipitation, DEM and NDVI.
- The defined neighbourhood is too small. Even if the first two cases do not occur globally, they may occur locally in the model.Therefore, the model results were tested several times by checking the local condition numbers in the output element classes(Oshan et al, 2020). A higher local condition number indicates that the results will be unstable due to local multicollinearity. If this occurs, we rerun the model using more neighbouring elements or distance ranges. As a general rule, results for elements with condition numbers greater than 30 or null are viewed with suspicion. When the condition number is scaled, the number of explanatory variables in the model is corrected. Finally, after several tests and corrections, we found that the condition numbers of the 10 explanatory variables retained were all less than 30.
References:
Breiman, L., and J. H. Friedman. 1985. Estimating optimal transformations for multiple regression and correlations (with discussion). Journal of the American Statistical Association 80, (391): 580–619. https://doi.org/10.2307/2288473. JSTOR 2288473.
Brunsdon C.A., S. Fotheringham, and M. E. Charlton. 1996. Geographically weighted regression: A method for exploring spatial nonstationarity. Geographical Analysis 28: 281–298.
Fotheringham, A. S., W. Yang, and W. Kang. 2017. Multiscale geographically weighted regression (MGWR). Annals of the American Association of Geographers 107: 1247–265. https://doi.org/10.1080/24694452.2017.1352480
Oshan, T. M., Z. Li, W. Kang, L. J. Wolf, and A. S. Fotheringham. 2019. mgwr: A Python implementation of multiscale geographically weighted regression for investigating process spatial heterogeneity and scale. ISPRS International Journal of Geo-Information 8: 269.
Yu, H., A. S. Fotheringham, Z. Li, T. Oshan, W. Kang, and L. J. Wolf. 2020. Inference in multiscale geographically weighted regression. Geographical Analysis 52: 87–106.
Thanks again for handling our work. If you have any questions, please do not hesitate to contact us.
Sincerely yours,
Cheng He & Huan He
Reviewer 2 Report
In this manuscript, Yao & Ma et al explored the migration process and influencing factors of O3 pollution based on measured data from urban monitoring sites in mainland China from 2015 to 2020. O3 pollution trend and the influencing factors during the period in different range of mainland China were reported. The manuscript was well prepared.
Comments
1, Check eq. 1 to 9, make sure every variable is defined and they are correctly. A few of them look strange, such as that in line 139 and eq. 8.
2, Data Collection, line 92-101
The negative and missing values were excluded. How about “abnormal” spike high concentrations? For example, a data set reading from a monitoring station were around 50. Then at some point, a spike high concentration reading of 200 was showed in the record, then back to around 50 again immediately. Did this kind of “abnormal” spike high concentrations exist in any of the data sets from any monitoring station? If yes, how did the authors handle those “abnormal” spike high concentrations?
3, Table 1 and Fig 1
In table 1, some min numbers are less than 40. Why the min number for the heatmap (Fig 1) is 40? For Dec. 2018 in Fig 1, what number of the “gray” color represents?
4, Fig 3, 4 and 5
The data for Fig 3, 4, and 5 were from the cities in each regain. It does not mean the whole region is the same. Authors should point it out in the text.
5, The data were analyzed for “regions”. How about the trend for a specific urban area or a specific monitoring station? What kind of hourly and daily variation they have? Does O3 concentration peaks at the same time on a given day as other pollution factors or there is a shift? It would be better to discuss about it.
6, Related to comment #5, regionally O3 concentration may not well correlated with a given pollution factors or a group of pollution factors. How about for a specific urban area or a given monitoring station? Does such correlation exist significantly? Or the O3 concentration may be correlated with a delayed pollution factor, such as O3 concentration may be correlated to PM2.5 concentration at one hour early? Did authors looked at such kind of correlations?
Author Response
Dear Reviewer,
Thank you very much for handling our submission and the constructive comments from the referees for our manuscript No. ijerph-2146763.
All the questions raised by the reviewers were addressed in the revised manuscript with highlights.
Response to Reviewers
1, Check eq. 1 to 9, make sure every variable is defined and they are correctly. A few of them look strange, such as that in line 139 and eq. 8.
Response: Thanks to the reviewer for pointing out the error. I have carefully checked the manuscript and corrected the wrong content.
2, Data Collection, line 92-101. The negative and missing values were excluded. How about “abnormal” spike high concentrations? For example, a data set reading from a monitoring station were around 50. Then at some point, a spike high concentration reading of 200 was showed in the record, then back to around 50 again immediately. Did this kind of “abnormal” spike high concentrations exist in any of the data sets from any monitoring station? If yes, how did the authors handle those “abnormal” spike high concentrations?
Response: Thank you very much to the reviewer for pointing out the error in the manuscript. The Ambient Air Quality Standard (GB3095-2012) sets out the minimum requirements for the statistical validity of O3 concentration data. 1-hour average concentration data for O3 should contain at least 45 minutes of valid sampling time, and 8-hour average concentration data for O3 should contain at least 6 hours of average concentration values. When the concentration evaluation items are counted at the city scale, all valid city monitoring points must be involved in the counting and evaluation. When the number of valid monitoring points is less than 75% of the total number of city points (less than 50% if the number of city points is less than 4), the statistics are considered invalid. 17 MDA8 data should exist from 08:00-24:00 on that day, of which at least 14 valid data should exist. If less than 14 data exist, but the MDA8 value has exceeded the limit value standard, the data statistics are also considered valid. If 324 or more valid MDA8 data exist in a calendar year and 27 or more valid MDA8 data exist in each month (25 in February), the statistics are considered to meet the validity requirements. In addition, we have also taken into account the adverse effects of anomalous values at individual monitoring stations at a given moment in time. For this reason, we counted 8-hour average concentration data for O3 containing the following two criteria: 1) at least 6 hours of average concentration values were included; 2) The difference between the individual moments of the same monitoring point does not exceed one times the minimum value over a period of six consecutive periods. Monitoring data that did not meet this requirement were excluded.
The specific details are shown on page 3, line 104-112.
3.Table 1 and Fig 1. In table 1, some min numbers are less than 40. Why the min number for the heatmap (Fig 1) is 40? For Dec. 2018 in Fig 1, what number of the “gray” color represents?
Response: Thanks to the reviewer’s suggestions. In table 1, the O3 concentration data are actual measurements containing 1425-1579 active monitoring stations; in Figure 1, the O3 concentration data are city average O3 concentration data based on 336 cities. Therefore, individual smaller or larger monitoring data in cities are averaged. In addition, the "grey" colour in Figure 1 represents the days when there were less than 14 valid MDA8 data and the MDA8 value did not exceed the limit value criteria between 08:00 and 24:00. Missing dates are 22-26 December 2018. Additional notes on this are given in the manuscript.
The specific details are shown on page7, line 249-251.
4, Fig 3, 4 and 5. The data for Fig 3, 4, and 5 were from the cities in each regain. It does not mean the whole region is the same. Authors should point it out in the text.
Response: Thanks to reviewer for pointing out the problems in this manuscript. Added labelling of data sources in Figures 3, 4 and 5.
The specific details are shown on page9,11-12 line 293-294, 333-334 and 355-356.
5, The data were analyzed for “regions”. How about the trend for a specific urban area or a specific monitoring station? What kind of hourly and daily variation they have? Does O3 concentration peaks at the same time on a given day as other pollution factors or there is a shift? It would be better to discuss about it.
Response: Thanks for the reviewer's comment. In order to represent the variation of urban O3 concentrations over different regions, three of the most polluted regions in eastern mainland China (Northern China(NC), Yangtze River Delta(YRD) region and Sichuan Basin(SB)) were selected for statistica analysis. From 2015 to 2020, the 90th percentiles of MDA8 in cities in NC, YRD and SB was 172±29 µg/m3, 160±23 µg/m3and 143±7 µg/m3, respectively. In mainland China, the overall trend of ozone concentration increase in the northern region(NC) is higher than the southern region(YRD and SB). In terms of inter-annual variation, the O3 concentration in cities in the four eastern regions of mainland China showed an upward trend from 2015 to 2018, and a downward trend after 2018.
Table 1 Statistical data of the 90th percentile of the MDA8 from urban monitoring stations in Northern China, the Yangtze River Delta and
the Sichuan Basin (2015-2020) (µg/m3)#
|
Region |
Year |
Counts |
Min |
Max |
Mean |
Coefficient of variation (%) |
|
Northern China |
2015 |
161 |
62 |
218 |
156±31 |
20 |
|
2016 |
153 |
91 |
217 |
159±24 |
15 |
|
|
2017 |
148 |
52 |
236 |
184±28 |
15 |
|
|
2018 |
158 |
111 |
230 |
187±22 |
12 |
|
|
2019 |
171 |
97 |
225 |
177±28 |
16 |
|
|
2020 |
158 |
75 |
214 |
166±25 |
15 |
|
|
Yangtze River Delta |
2015 |
195 |
52 |
203 |
150±33 |
22 |
|
2016 |
189 |
86 |
201 |
158±21 |
13 |
|
|
2017 |
175 |
114 |
211 |
171±20 |
12 |
|
|
2018 |
197 |
90 |
206 |
170±19 |
11 |
|
|
2019 |
214 |
118 |
192 |
161±18 |
11 |
|
|
2020 |
207 |
76 |
185 |
152±18 |
12 |
|
|
Sichuan Basin |
2015 |
96 |
56 |
200 |
140±28 |
20 |
|
2016 |
94 |
67 |
193 |
144±25 |
17 |
|
|
2017 |
96 |
60 |
184 |
149±20 |
13 |
|
|
2018 |
96 |
115 |
191 |
154±17 |
11 |
|
|
2019 |
102 |
97 |
205 |
141±19 |
13 |
|
|
2020 |
102 |
75 |
189 |
139±23 |
17 |
# MDA8, The annual mean O3 concentration of the maximum daily 8h average
Seasonal effects (a-d), monthly variation characteristics(c) ,weekend effects(d) and day-night differences(b) of O3 concentration in the three eastern regions of mainland China from 2015 to 2020 year are show in Figure1-3.The daily peak of urban O3 concentration in the three eastern regions of mainland China appeared around 15:00-16:00, and the trough value around 8:00-9:00(CST (China Standard Time, UTC+0800)). However, the daily peak of urban O3 concentrations occur slightly earlier in the northern regions (NC) of mainland China than in the southern regions (YRD and SB). This may be due to a faster rise in temperature in the northern regions.
Fig. 1 Seasonal effects(a-d), monthly variation characteristics(c) ,weekend effects(d) and day-night differences(b) of O3 concentration in northern China from 2015 to 2020 year.
Fig. 2 Seasonal effects(a-d), monthly variation characteristics(c), weekend effects(d) and day-night differences(b) of O3 concentration in the Yangtze River Delta , China from 2015 to 2020
Fig. 3 Seasonal effects(a-d), monthly variation characteristics(c) ,weekend effects(d) and day-night differences(b) of O3 concentration in the Sichuan Basin, China from 2015 to 2020
The Yangtze River Delta region was used as a case study where the trends of six environmental pollutants (O3, PM2.5, PM10, CO, NO2 and SO2) on time scales were significantly different. On a seasonal scale, ozone concentrations are highest in summer and the other five pollutants (PM2.5, PM10, CO, NO2 and SO2) are highest in winter. On a daily scale, the daily peak of urban O3 concentrations occur at 15:00-16:00 (Fig. 4a), while peak CO concentrations occur at around 7:00-8:00 (Fig. 4d).The first peak of PM2.5 and PM10 occurs around 8:00-9:00 and the second peak starts around 22:00 (Fig. 4c; Fig. 4d). SO2 and NO2 show opposite trends, with peak concentrations occurring at around 09:00 and 19:00 respectively (Fig. 4e; Fig. 4f).
Fig. 4 Day-night differences of O3, PM2.5, PM10, CO, NO2 and SO2 concentration in the Yangtze River Delta region, China from 2015 to 2020(a: O3; b: PM2.5; c: PM10; d:CO; e:NO2; f: SO2)
6, Related to comment #5, regionally O3 concentration may not well correlated with a given pollution factors or a group of pollution factors. How about for a specific urban area or a given monitoring station? Does such correlation exist significantly? Or the O3 concentration may be correlated with a delayed pollution factor, such as O3 concentration may be correlated to PM2.5 concentration at one hour early? Did authors looked at such kind of correlations?
Response: Thanks to the reviewer’s suggestions. On an hourly scale, we explored six pollutants (O3 concentration with PM2.5, PM10, CO, NO2 and SO2) in cities in the Yangtze River Delta region in 2018. The results showed that urban O3 concentration are highly significantly negatively correlated with PM2.5, PM10, CO, NO2 and SO2 in in the Yangtze River Delta region in 2018 (Fig. 5). In addition, we found a particularly interesting finding that urban O3 concentration are highly significantly negatively correlated with PM2.5 concentration at 1, 6, 12, 18, 24, 48 and 72 hours in advance, and this negative correlation became more pronounced with increasing time. Once again, we thank the reviewers for providing new directions for our future research. This will be our research in more depth (Fig.6).
Fig.5 Correlation analysis of urban O3 concentration with PM2.5, PM10, CO, NO2 and SO2 in the Yangtze River Delta region in 2018.
Fig.6 Correlation analysis of O3 concentrations with PM2.5 at 1, 6, 12, 18, 24, 48 and 72 hours in advance in the Yangtze River Delta region in 2018.(Note: Adv01, Adv06, Adv012, Adv018, Adv24, Adv48 and Adv72 represent urban PM2.5 concentration at 1, 6, 12, 18, 24, 48 and 72 hours in advance, respectively.)
The specific details are shown on page14, line 452-456.
Thanks again for handling our work. If you have any questions, please do not hesitate to contact us.
Sincerely yours,
Cheng He & Huan He

Reviewer 3 Report
The manuscript talked about the temporal and spatial distribution of atmospheric surface O3 in China. The authors investigated the influencing factors of the O3 distribution in China with different statistical methods. Generally, the results of the study are interesting and important for the air quality study especially in China. But the manuscript is too much in result description, the depth of discussion should be improved. Some comments and suggestions: 1. line 30, what is the "gravity center of the urban O3"? And, what is the importance of the gravity center? Corresponding information was not found in the text. 2. Please give the definition of DEM? 3. When discussing the factors influencing the distribution characteristics of urban O3 concentration, I would suggest to analyze the impact of population (or population density) and GPD value, quantitatively. 4. line 359-362, the authors fount that SO2, PM2.5, DEM, sunshine hours, and precipitation were mainly positively correlated with the spatial distribution of O3, while the NO2 concentration and NDVI were negatively correlated with the spatial distribution of O3. what are the reasons for the correlations found? Does the type of vegetation have some impact on the O3 distribution? The authors should have more discussions. 5. line 394-395, NOx and VOCs could not be the main components of PM2.5. Please modify. Actually, I'm quite interested to know the positive correlation of PM2.5 and O3 distribution. Do such correlation exist in all seasons? How to understand the mutual influence relationship of PM2.5 and O3? Could the authors talk more on the correlation between PM2.5 and O3? 6. Section 3.5, 21 combination were listed in Figure 7 and it is hard to see clearly. I would suggest to focus on the really interesting ones and have more discussion to show the importance of the findings. Some of the figures could be put into supporting information. This is also true for the other sections.
Author Response
Response to Reviewers
Dear Reviewer,
Thank you very much for handling our submission and the constructive comments from the referees for our manuscript No. ijerph-2146763.
All the questions raised by the reviewers were addressed in the revised manuscript with highlights. Our itemized responses are also presented below in this reply letter.
- line 30, what is the "gravity center of the urban O3"? And, what is the importance of the gravity center? Corresponding information was not found in the text
Response: Thanks for the reviewer's comment. I am very sorry for not explaining this clearly. The gravity center of urban O3 refers to the center of O3 movement, which represents the central location where high O3 concentration are concentrated in mainland China during 2015-2020.
- Please give the definition of DEM.
Response: Thanks to the reviewers for pointing out the problems with our manuscript. DEM stands for Digital Elevation Model, the meaning of which has been completed in the manuscript.
The specific details are shown on page13, line 380.
- When discussing the factors influencing the distribution characteristics of urban O3 concentration, I would suggest to analyze the impact of population (or population density) and GDP value, quantitatively.
Response: Thank the reviewers for their valuable comments. We chose 2018, the most polluted year for urban O3 in mainland China, as the year of study for the relevant analysis. The results show a highly significant positive correlation between urban O3 concentration and population density and GDP per capita in mainland China in 2018 (Fig.1). Among them, the correlation between population density and urban O3 concentrations is higher, indicating that urban O3 concentrations are higher in densely populated areas.
Fig.1 Correlation analysis of O3 concentrations with population density and GDP per capita in mainland China in 2018. (Note: PD: population density; GDP: GDP per capita)
- line 359-362, the authors fount that SO2, PM5, DEM, sunshine hours, and precipitation were mainly positively correlated with the spatial distribution of O3, while the NO2 concentration and NDVI were negatively correlated with the spatial distribution of O3. what are the reasons for the correlations found? Does the type of vegetation have some impact on the O3 distribution? The authors should have more discussions.
Response: Thanks to the reviewers for their comments. In response to the reviewers' suggestions, we have made revisions in the manuscript, which are shown on page 13. The revisions are shown below.
“Emissions of atmospheric gas pollutants may contribute to urban O3 formation, and the precursors of secondary particulate matter in PM2.5, which also contains ozone precursors, can contribute to urban O3 concentrations. There is a certain relationship between the photochemical oxidation of SO2 and O3. A high O3 concentration greatly promotes the chemical conversion of SO2 to sulfate (Wang et al., 2019). Meteorological factors play an important role in influencing near-surface ozone concentrations, but the dominant factors show different effects on ozone in different regions (Xu and Zhu, 1994). The duration of light and the amount of evaporation are related to solar radiation. Intense solar radiation (sunshine hours) drives active photochemical reactions, which increase urban ozone concentrations. Similarly, solar radiation is stronger at higher altitudes than at lower altitudes due to the weakening effect of clouds and water vapor at lower altitudes that exacerbate the solar radiation. In addition, the effect of vegetation on regional O3 pollution is also very closely related. Studies have shown that during drought, vegetation is less effective in removing O3 through stomata than during wet periods, exacerbating the extreme effects of O3 pollution (Lin et al., 2020). ”
The specific details are shown on page13, line 386-400.
In addition, in response to the question “Does the type of vegetation have some impact on the O3 distribution? ”. Many thanks to the reviewers for their suggestions. We have not classified and extracted the vegetation types yet, but we will focus on this part in detail in our later research.
References:
Wang, Z.B., Li, J.X., Liang, L.W. Spatio-temporal evolution of ozone pollution and its influencing factors in the Beijing-Tianjin-Hebei Urban Agglomeration. Environ. Pollut. 2020, 256: 113419
Xu, J.L., Zhu, Y.X., Effects of the meteorological factors on the ozone pollution near the ground. Chin. J. Atmosph. Sci. 1994, 18, 751–757.
Lin, M Y., Horowitz, L W., Xie, Y Y., Paulot, F., Malyshev, S., Shevliakova, E., Finco, A., Gerosa, G., Kubistin, D., Pilegaard, K. Vegetation feedbacks during drought exacerbate ozone air pollution extremes in Europe. Nat. Clim. Change. 2020, 10(5): 444-451.
- line 394-395, NOx and VOCs could not be the main components of PM5. Please modify. Actually, I'm quite interested to know the positive correlation of PM2.5 and O3 distribution. Do such correlation exist in all seasons? How to understand the mutual influence relationship of PM2.5 and O3? Could the authors talk more on the correlation between PM2.5 and O3?
Response: Thanks to the reviewers for their valuable suggestions. We revised the misrepresentation in the original manuscript. The main component of PM2.5 was composed of gaseous pollutants (such as carbon, organic carbon compound, sulfate, nitrate etc.)
The correlation between PM2.5 and O3 concentration in different seasons was further analyzed. Under the effect of different meteorological elements, the influence of PM2.5 concentration on O3 concentration was a complex process, however, urban PM2.5 and O3 concentrations showed a significant negative correlation in all seasons on the time scale. A significant negative correlation (r = -0.423) was found between O3 and PM2.5 in mainland Chinese cities from 2015-2020. Within the 99% confidence interval, the correlation coefficients between urban O3 and PM2.5 in different seasons were -0.293 (spring), -0.080 (summer), -0.272 (autumn) and -0.296 (winter). These phenomena have been attributed to the fact that high concentrations of particles lead to an increase in the optical thickness of the aerosol, which weakens the photochemical production rate of ozone and contributes to a decrease in ozone concentration. In addition, an increase in PM2.5 concentration could weaken atmospheric radiation and thus suppress ozone levels through the disappearance of UV light (Wang et al., 2020).
The specific details are shown on page 14, line 440-452.
Reference:
Wang, Z.B., Li, J.X., Liang, L.W. Spatio-temporal evolution of ozone pollution and its influencing factors in the Beijing-Tianjin-Hebei Urban Agglomeration. Environ. Pollut. 2020, 256: 113419
- Section 3.5, 21 combination were listed in Figure 7 and it is hard to see clearly. I would suggest to focus on the really interesting ones and have more discussion to show the importance of the findings. Some of the figures could be put into supporting information. This is also true for the other sections.
Response: In the GAM model, the F-values is used in statistical tests of models, and the higher the F number, the more important the index. Therefore, We selected six interaction terms with large F values for analysis (Fig.2).
Fig.2 Interaction of different factors on the variation of urban O3 concentration Three-dimensional effect plots. a. NO2 and sunshine hours (SH); b. SO2 and sunshine hours (SH); c. PM2.5 and sunshine hours (SH)ï¼›d. Digital Elevation Model (DEM) and sunshine hours (SH); e. precipitation (PREC) and sunshine hours (SH); f. normalized difference vegetation index (NDVI) and sunshine hours (SH).
Thanks again for handling our work. If you have any questions, please do not hesitate to contact us.
Sincerely yours,
Cheng He & Huan He
Round 2
Reviewer 1 Report
Separation of scales is necessary for multivariate analysis to avoid erroneous results. High frequency noisy component must be removed to avoid falsifications in coefficients of influences.
Author Response
Dear Reviewer,
Thank you very much for handling our submission and the constructive comments from the referees for our manuscript No. ijerph-2146763.
All the questions raised by the reviewers were addressed in the revised manuscript with highlights. Our itemized responses are also presented below in this reply letter.

Reviewer 3 Report
The authors responded to the comments raised by the reviewers. The revised manuscript is better than the original version, but I think the authors should still try to improve the discussion to show more clearly the reasons behind the statistical results and help the readers to understand more easily the situation of O3 pollution in China. One specific comment is for the sentence in line 417-418. I guess the authors intend to say that some of the components in PM2.5 such as nitrate and secondary organic carbon are also formed from gaseous pollutants of NOx and VOCs?
Author Response

(The authors gave the same response as above.)
